# Water Recognition on the Moon by Using THz Heterodyne-Spectrometer for Identifying the Appropriate Locations to Extract Water for Providing Oxygen for Breathing and Fuel for Spaceships' Propulsion on the Moon with CubeSat

**Vahid Rastinasab** * , **Weidong Hu and Mohammad Kazem Tahmasebi**

Department of Electronic and Information, Beijing Institute of Technology, 5 Zhongguancun St, Haidian Qu, Beijing 100811, China; hoowind@bit.edu.cn (W.H.); mkts1514@gmail.com (M.K.T.)
* Correspondence: vahid.rastinasab@gmail.com

**Abstract:** Asteroid mining offers vital sources for improving human lives and provides opportunities for interplanetary missions and space travel. There are many professional commercial space companies that are only investing billions of dollars on asteroids mining, but prior to that, one condition for asteroid mining could be planetary stations to refuel the pioneers' spacecraft or human colonies on alien planets; hence, one of the vital sources for these purposes is water. Water can be harvested to split oxygen for breathing and hydrogen for refueling spaceships' propulsions, and Earth-to-space water payload transporting is extremely expensive; therefore, discovering extraterrestrial water in outer space is economically beneficial. This paper presents a Lunar CubeSat Injector to deliver four 3U CubeSats into Low Lunar Orbit to make a constellation to identify locations of water sources on the Moon by using a THz heterodyne-spectrometer. In sum, this project can help scientists to recognize more water resources for those who will colonize the Moon and for those planning to go beyond it.

**Keywords:** deep space prospecting; asteroids mining; THz remote sensing; water detection by spectrometer

## 1. Introduction

There are millions of asteroids in our solar system, comprising a larger source of minerals than what is present on Earth. Mining asteroids for precious materials such as platinum, gold, and iridium, will provide opportunities to generate new alloys and form new businesses. The next generation of trillionaires comprises asteroids miners; subsequently, it will attract many commercial space companies to invest in asteroid mining and develop present technologies. With existing technology, harvesting of asteroids is not affordable compared to its benefits [1]. Therefore, some preparations must be done before starting to mine asteroids. One interesting idea is establishing an asteroid miner station near those asteroids or on the Moon. What makes extraterrestrial miner stations constructible is water. Nothing is more crucial than water. Water serves multiple functions, which include life support, hydrogen to refuel propellant, thermal shield, and agriculture [2]. When water sources are detected, for example, on an alien body such as the Moon, the extraterrestrial miner stations could be constructed near the water sources, and then pioneers robots could harvest the asteroids and bring them into the station. The proposed idea is not possible in the near term, but it forms the main intention of this paper and the reason for sensing the location of water on the Moon. NASA is partnering with commercial space companies such as Blue Origin, SpaceX, and Astrobotic to develop robotic and lunar rover missions. Jeff Bezos and Blue Origin announced the target of establishing a habitable lunar station near lunar poles where astronomers can live and work [2]. The Moon is the next generation of future human settlement for interplanetary missions and

prospecting of our solar system. There are millions of liters of ice water on the Moon's surface [3,4]. Water on the Moon's surface can provide oxygen for breathing, fuel for propellants, and life support; therefore, water is a fundamental need when it comes to living on alien planets, asteroids, natural satellites, etc. This precious liquid has a very good sign at the THz frequency range. THz passive remote sensing technology has expanded with the development of electronic components, especially Complementary metal–oxide–semiconductor (CMOS) and Schottky diodes. THz, with its short wavelengths, are adapted for appropriate detecting the small particles of water vapor and ice water. NASA has recently been developing a 90/180 GHz radiometer hurricane mission. The radiometer contains a fixed beam with dual canaliculated horns with a lens antenna to operate at 90 and 180 GHz frequency range for remote sensing of soil moisture [5,6]. THz's application for spectroscopy by analyzing the spreading of THz wavelength through the water vapor has achieved a new level of high brightness, making is the most accurate model to absorb across realms of water vapor in the frequency range of 0.2 to 1.45 THz [7]. Reference [8] has two goals: firstly, to compare the amount of perceptible water vapor with ground-based scanning radiometer with atmospheric radiation measurement, and secondly, to compare the radiometric response of THz and microwave wavelengths to ice cloud and water. A high-performance THz receiver for monitoring Jupiter's ice moon has been developed by the Chalmers University of Technology. Remote sensing of vapor and gases by high-quality heterodyne receivers has been developed for monitoring planetary missions [8]. The permittivity of water in the THz range enables the detection and imaging of soil moisture capability. This property is utilized to make a laboratory experience for remote sensing and mapping soil water moisture by a THz radiometer. Examples of THz scanning for foods, agricultural productions, and polymers are reviewed in [9]. Low-power, ultrashort THz pulses that radiate in the laboratory environment have enabled the capability to detect signatures of water vapors [10]. An interesting method to determine snow ice water, liquid water path, and integrated water vapor has been done by a neural network algorithm based on an airborne millimeter and sub-millimeter radiometer [11]. MicroMAS a joined project of MIT/LL (Massachusetts Institute of Technology/Lincoln Laboratory), a 3U CubeSat hosting a passive micro-sized spectrometer near 118.75 GHz frequency has been used to observe the formation process of tropical cyclones, which can determine cloud formation from small particles of water to a storm [12]. MiRaTA CubeSat mission of MIT/LL (Massachusetts Institute of Technology/Lincoln Laboratory) contains a constellation mission that was launched in early 2016 with tri-band radiometer at frequencies (60, 183, and 206 GHz) for real-time remote sensing of Earth's atmospheric [13]. Submillimeter Wave Astronomy Satellite (SWAS) with small satellites have been used to determine water vapor on Jupiter and Saturn with multiband THz radiometry at center frequencies 489 and 553 GHz, and experimental data have shown that THz radiometer performed 1.5–2.5 times better than Infrared Space Observatory (ISO) for water vapor signs, showing that water has better signs at THz waves [14].

In addition to theoretical concepts, the available hardware technologies must be considered to verify the feasibility of manufacturing the THz heterodyne spectrometer. CubeSat system specifications present some constraints in size, mass, power generation, etc. Hence, the THz heterodyne spectrometer requires high precision in design. In [15], a high-performance, low-power-consumption, integrated THz Schottky receiver is presented for CubeSats, which is a multilayer receiver with approximate dimensions of $25 \times 25 \times 40$ mm and mass less than 250 g. This multi-frequency receiver operates at 557 GHz and 1100–1200 GHz. These frequency ranges provide an excellent indication of ice water [16].

The previous paragraphs show that THz technology has attracted great attention, particularly in establishing their hardware on small satellites due to short THz wavelengths. In this paper, we propose a scenario to discover water on the Moon by using THz heterodyne-spectrometer by using a 3U CubeSat. This scenario includes three main segments: first, a Cargo Microsatellite to transport four CubeSats into Low Lunar Orbit (LLO); second, four CubeSats for lunar missions, which utilize a gas thruster in its Attitude Control System

where the magnetic-based Atitute Determination and Control Subsystem (ADCS) due to low lunar gravity are useless; Third, The THz heterodyne-spectrometer payload, which has new characteristics in its antennas specifications. A $3 \times 3$ array horn antenna interface with CMOS switchers instead to patch array antennas. Horn antennas generally provide more gain compared with patch array antennas, and we can still use options of an array antenna. More details are explained in the next sections. In the paper, firstly, we present mission objectives, then continue on mission scenario, and at the end, we present the satellite project system with a THz radiometer for detecting ice water on the Moon for creating the next generation of deep space exploration by using small satellites [17–22].

## 2. Mission Objective

Water is a vital source around the globe. Water is required for breathing, planting, and agriculture, and all life depends on it. The Moon has attracted the greatest attention of NASA, SpaceX, and most of the professional commercial space companies and investors because the Moon is a window into human space exploration. Scientists have discovered that there are millions of liters of ice water on the Moon's poles [23]. Liquid water cannot survive on the surface. Water quickly reacts chemically and turns into steam, and water vapor goes to outer space. Therefore, the water on the Moon only exists in ice water form. Some researchers have acknowledged that THz short wavelengths have a high potential to identify water particles.

A CubeSat-sized THz heterodyne-spectrometer has some constraints, such as mass, power, and dimension. NASA's JPL has already developed a CubeSat-sized THz spectrometer for detection of water on Jupiter and Saturn, and there are also three most updated THz CubeSats that they operated for Earth's atmosphere monitoring. Previous missions assist us to develop the multi-channels THz spectrometer for determining ice water on the Moon. This technology is a low-power consumer of CMOS or Schottky diodes based on THz receiver systems [24–26].

For this mission, a cargo Microsatellite carries four 3U CubeSats into Low Lunar Orbit and injects them into Low Lunar Orbit. After this, these five satellites make a constellation to identify ice water locations on the Moon surface via a THz spectrometer. The Cargo Microsatellite receives data from CubeSats and transmits them to Earth Deep Space Network (DSN). Meanwhile, we already have LUMIO built by European Space Agency [27] and MarCO built by NASA Jet Propulsion laboratory [28], CubeSats for prospecting of the Moon and Mars, respectively, but our project is proposing the followed prospective beneficiaries: (1) using CubeSats for deep space missions, (2) developing the CubeSat constellation for interplanetary missions, and (3) providing return samples capability from other planets.

## 3. Lunar Orbit CubeSat Injector Scenario

The Lunar Orbit CubeSat Injector (LOCI) is a Cargo Microsatellite to transport four third-party 3U CubeSats into Low Lunar Orbit (LLO). The LOCI must contain a gas thruster (the model is Aerojet MPS-130 thruster) and four Poly Picosatellite Orbital Deployers (P-POD) as the key requirements to transport the CubeSats from Earth orbit to LLO [29]. Additional explanations about system-level design are described in Section 4. The trajectory route to LLO orbit requires four main thrusts series (see Figure 1):

1. Thrust near Geostationary Transfer Orbit (GTO)perigee to increase Geostationary Orbit (GEO) orbit radius altitude.
2. Thrust near apogee to increase the radius of GTO perigee to ensure lunar orbit transferring.
3. Thrust and spiral to capture a lunar orbit.
4. Thrust to enter 100 km altitude circular lunar orbit.

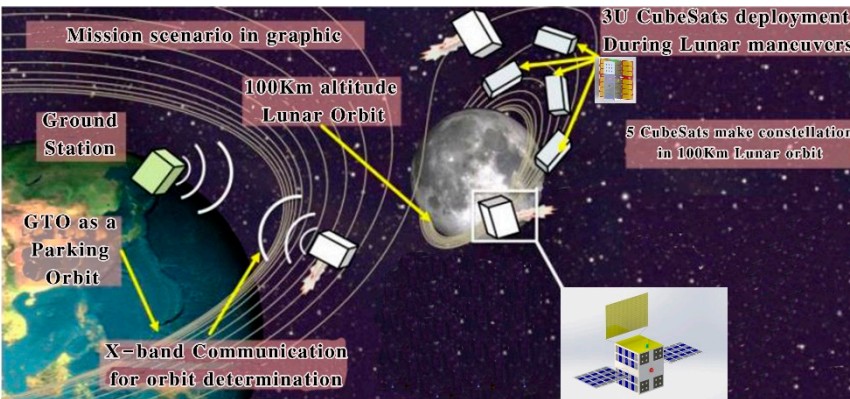

**Figure 1.** LOCI concepts of operation.

Figure 1 shows LOCI's scenario. After this, LOCI arrives 100 km LLO, the four third-party CubeSats enters into the orbit; at this time, LOCI has a camera in the front to take a photo of CubeSats injection from P-PODs and transmits the image by X-band transmitter to announce Earth station, which the CubeSats have been successfully inserted into LLO. Figure 2 shows CubeSat modes. After the deployment phase, the CubeSat enters into Safe mode, at this mode the CubeSat actives Electronic Power Subsystem (EPS) and On Board Data Handling Subsystem (OBDH) and wait until 15 min for batteries to be fully charged and for deployable mechanisms such as UHF antennas and solar cells spread properly. Then, the CubeSat sends a beacon and enters Recovery mode. In Recovery mode, CubeSat Attitude Determination and Control Subsystem (ADCS) adjusts the CubeSat's orientation to nadir pointing attitude (the payload on the CubeSat operates only when orients to the Moon's surface). After Recovery mode is properly accomplished, the CubeSat transmits a beacon to the Microsatellite and enters Normal mode (the next mode). In Normal mode, the CubeSat waits for Ground Station (GS) commands to enter three functional modes, Constellation mode, Communication mode, and Payload mode. In all the modes, the Microsatellite communicates data relay between CubeSats and Ground Station. Payload mode means the THz heterodyne-spectrometer turns ON and begins to collect lunar surface data, then saves the data on a 32 GB Secure Digital (SD) card. When the SD becomes full, the CubeSat enters Constellation mode to transmit the payload data. Constellation mode means either a new command has been received from Ground Station, and the Microsatellite spreads the received command to all CubeSats, or one of the CubeSats transmits the payload data to Microsatellite to relay data to Ground Station. From each mode, satellites can enter Safe mode; at Safe mode, all subsystems turn OFF, the batteries are charged, and OBDH goes into power-down mode. After each functional mode, the CubeSats return Normal mode and stay until it receives GS commands.

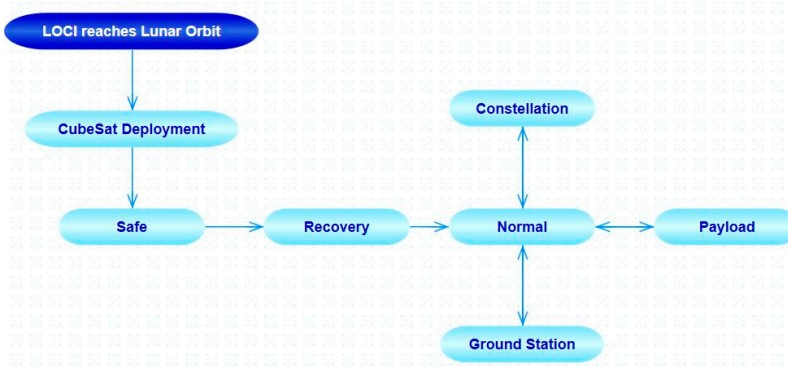

**Figure 2.** Mission Scenario.

## 4. System-Level Design

Lunar water is water that has been found on the Moon. The research shows that liquid water cannot survive on the Moon's surface. The water vapor is decomposed by sunlight, leaving oxygen and hydrogen lost to outer space. Scientists have conjectured that ice water can exist on the Moon, and there are millions of liter of ice water on the Moon's poles [30]. Here, we present a CubeSat THz heterodyne spectrometer for the identification of ice water on the Moon. This idea considers the capability of system-on-chip THz communication and millimeter receiver for space exploration and remote sensing of the Moon to detect water. The THz payload prepares technics and calibrations to provide water sources to increase the potential of humans surviving in the harsh environment of the Moon. In addition, the CubeSat platform plays a key role in accomplishing the mission. A CubeSat has limitations in mass, size, and electrical power. A general CubeSat, for example, a 3U CubeSat, has a maximum 10 kg mass subjected to Poly-Picosatellite Orbiter Deployer (P-POD) launcher system standard. A standard CubeSat platform includes Electrical Power Subsystem (EPS), On Board Data Handing (OBDH), Telemetry Tracking and Command (TTC), Attitude Determination and Control Subsystem (ADCS), Thermal Control Subsystem (TCS), Structure Design (Str), and thruster. The platform specifications depend on the mission requirements [31].

Part of this paper includes a description of a cargo Microsatellite (named LOCI). LOCI transports four third-body CubeSats into LLO. Meanwhile, many low-power consumer thrusters have been built to be compatible with CubeSats, but in our design, we prefer to utilize the CubeSats' thruster only for ADCS, where a gravity-based ADCS system is useless on the Moon. By using LOCI's thrusters, the trajectory time from the Earth to the Moon will be done more quickly. The paper mostly explains the CubeSat system, but LOCI's key subsystems are also illustrated.

### 4.1. Payload Design and Primary Sketch

Heterodyne spectrometers at millimeter and submillimeter frequency ranges are widely utilized in Earth and lanetary science to identify the origins of galaxies and to study the characteristics of Earth's surface and atmospheric compositions [32]. HIFI [33], MIRO [34], and MLS [35] are space-borne THz heterodyne-spectrometer examples that already launched into the Earth's orbit. These small satellites with millimeter spectrometry missions and their functionalities in the Earth's remote sensing missions provide the background for upgrading current terrestrial THz spectrometer technologies for outer space missions. In [35], a 183 GHz receiver with a tunable center frequency was manufactured for remote sensing of atmosphere.

We present a miniaturized THz heterodyne-spectrometer, which contains three $10 \times 10$ cm electronic boards with 3.15 W total power consumption and 1 kg total mass. The payload is compatible with the CubeSat platform standard; therefore, the payload operates on the CubeSat and senses the ice water footprints on the Moon's surface. NASA is developing a spectrometer system with a frequency 300 up to 4700 GHz. The 500–600 GHz frequency has responses for water, and the 509 to 590 GHz range has responses for water vapor; this receiver is a CMOS-based RF circuit. Additionally, THz Schottky diodes receivers present a high gain in performance, especially when cooled until $-100$ K. The THz Schottky receiver offers a CubeSat platform suitable for 1U CubeSat. There are many advantages of using THz receiver on small CubeSats. The motivation of this paper is to develop a millimeter receiver for water vapor and ice water detection on the Moon. Water vapor has an excellent response at frequency 557 GHz and in 1100–1200 GHz bands. A 557 GHz spectrometer was the main target of the NASA SWAS mission. The first block diagram for THz heterodyne-spectrometer is illustrated in Figure 3. It contains three main blocks, Horn antennas switchers, 28 nm CMOS heterodyne RX, and 65 nm CMOS Spectrometer to generate the spectrum out. Array horn antennas with $3 \times 3$ dimensions can provide higher gain, increase the directivity, and steer a radio beam to point at different orientationss or find the signal direction [36–38].

**9\*Horn antennas**

**Figure 3.** THz Heterodyne-Spectrometer block diagram.

References [39–42] confirm that the THz receiver for water detection is feasible with a CubeSat platform, and state-of-the-art technology shows that by using Schottky diodes or a CMOS-based receiver, we can implement the proposed idea. We propose a multi-channel THz receiver to determine water vapor, ice water, and liquid water; see Figure 4.

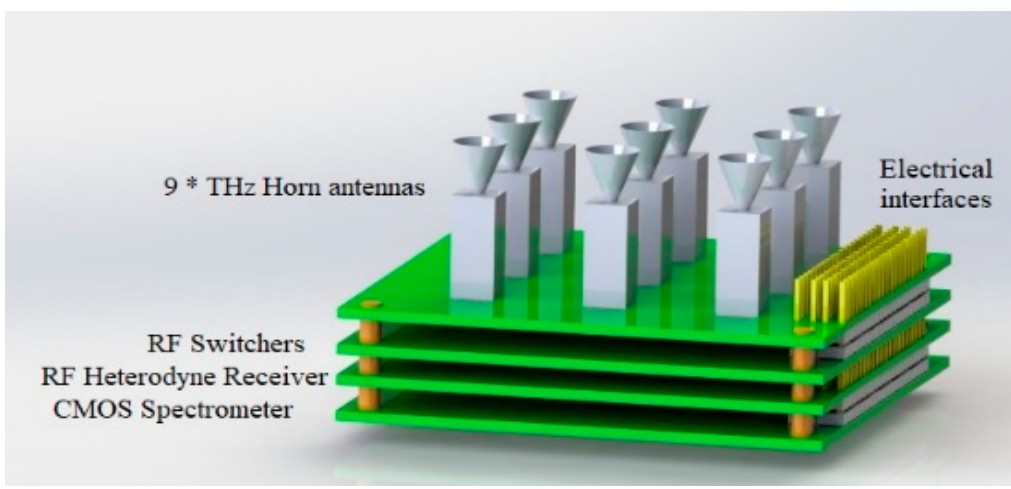

**Figure 4.** Multichannel THz heterodyne-spectrometer first sketch.

### 4.2. Lunar Orbit Insertion and Constellation

Three classes of orbits are considered for the project: libration orbit, perturbed Keplerian orbit, and Keplerian orbit. Only orbits with repetitive or periodic characteristics with respect to the Moon surface can be considered as remote sensing orbits of the Moon. To consider Keplerian orbits, we include Low Lunar Orbit (LLO) and Elliptical Lunar Orbit (ELO). LLO has a constant altitude with respect to the Moon surface and a lower orbit period (usually it would be 2 h with alt = 100 km). If the altitude > 100, the Moon gravity is not the only gravity that affects the satellite; the Earth's gravity causes the altitude to becomes variant in some positions. ELO typically have a larger apolune altitude and a lower perilune altitude; therefore, the spacecraft-to-the-Moon distance changes periodically in an orbital motion along with a certain coverage of lunar regions [43]. We have chosen LLO as CubeSat lunar remote sensing orbit (see Figure 5). The LOCI enters LLO and injects the CubeSats, and then the group of satellites (the four CubeSats and the cargo microsatellite) work together and form a lunar constellation.

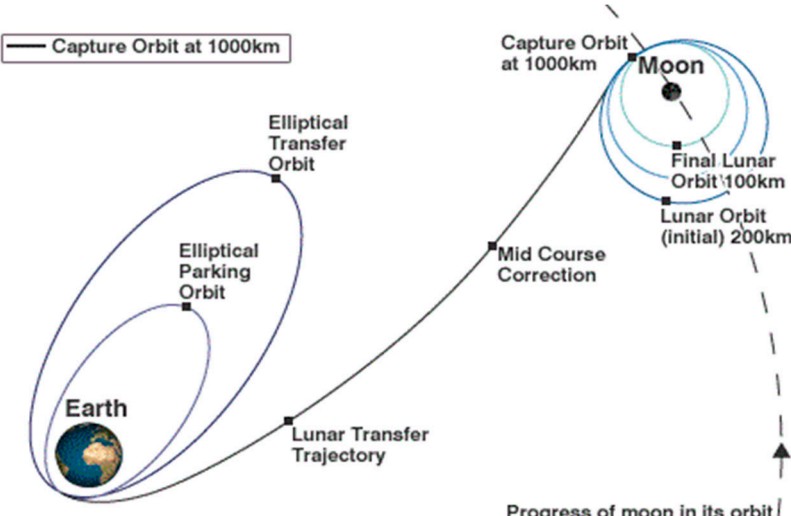

**Figure 5.** LOCI Low Lunar Orbit insertion.

LLO orbit insertion sequence contains several large maneuvers followed by a small thrust made by the LOCI thrusters. All maneuvers typically take 867 days to capture the LLO orbit with a 100 km altitude. Typically, each orbit maneuver takes 38 min using four LOCI thrusters. Due to several contingencies in the Earth-to-Moon trajectory, LOCI is planned to have such long orbit maneuvers to minimize the third-body effect on larger capture orbit. Reference [44] estimated time for lunar orbit capture with LOCI. At this stage, when the LOCI captures the lunar orbit, it deploys four CubeSats (see Figure 6) to reach different orbits for prospecting of the Moon for ice water by using their THz heterodyne spectrometers.

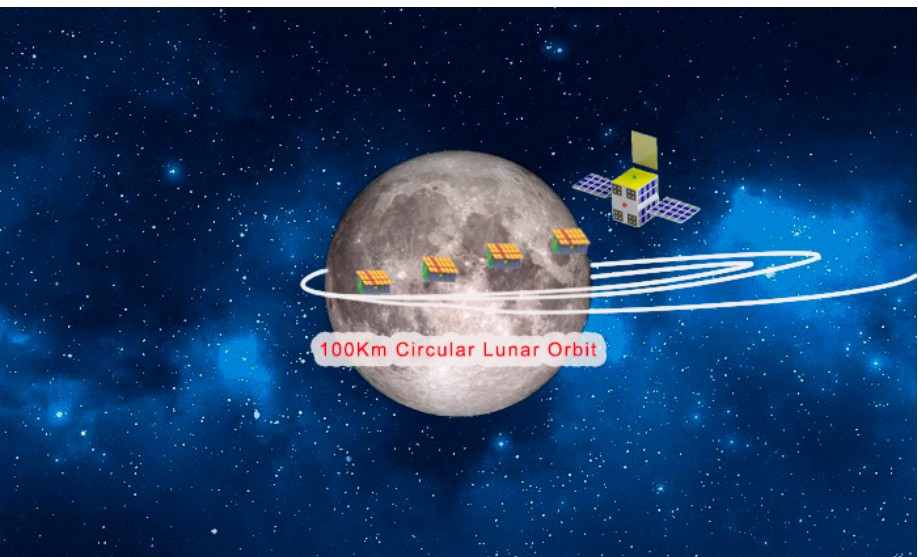

**Figure 6.** LOCI Low Lunar Orbit insertion.

### 4.3. Attitude Determination and Control Subsystem

Outer space environment properties present special characteristics that require a specific ADCS system. A magnetorquer-based ADSC system is useless in lunar conditions due to the low magnetic field around the Moon; hence, a dedicated ADCS system for a lunar mission is a Reaction-Wheel-based control system; however, the RW system still requires a gas thruster to desaturate the momentum management systems [44]. Table 1 shows LOCI's ADCS items list.

**Table 1.** LOCI's ADCS items list.

| Microsatellite Characteristics | | |
|---|---|---|
| **Components** | **Qty** | **Model** |
| sun sensor | 2 | SolarMEMS nanoSSOC-D60 |
| Star tracker | 2 | Hyperion ST400 |
| IMU | 1 | Sensonor STIM 300 |
| Thruster | 2 | Aerojet MPS-130 |
| Reaction Wheels | 3 | TBD |

LOCI requires high accuracy in Earth-to-Moon orbit trajectory and CubeSats orbit deployment; hence reaction wheels are required as part of LOCI's ADCS, but for the CubeSats' ADCS, the cold gas thruster is sufficient. The CubeSat Attitude Determination Control Systems are similar to LOCI components, except for two items, star tracker, and thruster. In [45], a cold gas thruster for interplanetary mission by UT Austin is developed; see Figure 7. The project is named INSPIRE. INSPIRE Attitude Direction Control only utilizes cold gas thruster. The star tracker that was selected is [46] item. A small satellite compact-sized star tracker. INSPIRE is a 3Unit CubeSat similar to our proposed system. The thruster only occupies $6.4 \times 9.0 \times 9.2$ cm, includes the circuit hardware, gas tank, and nozzles. The thruster computer power consumption is 0.2 W, and each nozzle consumes 5.2 W upon which it opens. This means when all four nozzles open, the power consumption is approximately 21 W, but since every thrust only operates in milliseconds, it is compatible with the power generation of a 3U CubeSat.

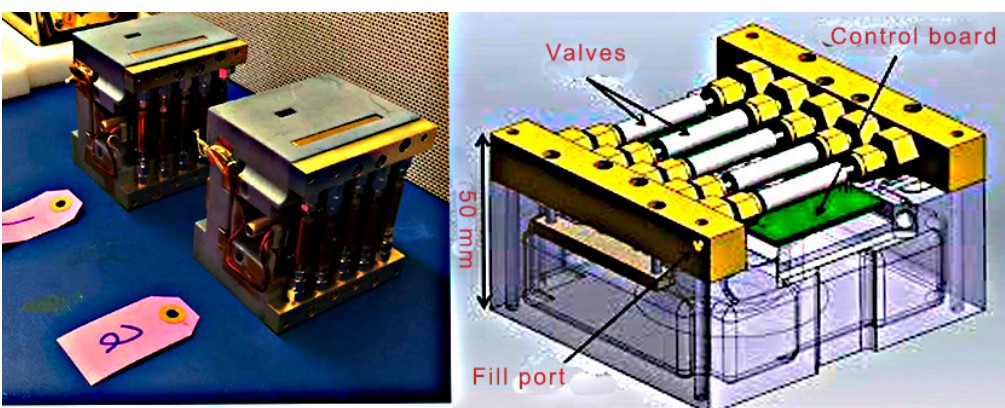

**Figure 7.** CubeSat cold gas thruster.

### 4.4. Telemetry Tracking and Command Subsystem

When a satellite launches into orbit, the only communication method is the TTC subsystem; hence, this subsystem is a crucial subsystem and must be chosen accurately. LOCI's TTC requirements are as follows: (1) the mother ship should be able to transfer all CubeSats data to the ground station in the shortest time possible; (2) the CubeSats must be able to find the mother ship (LOCI) without the need for pointing position; (3) CubeSats should have a supporting radio link with low bit rate when S-band link is faced with malfunctioning; (4) CubeSats must be able to transfer their data to the ground station if mother ship communication link is faced with malfunctioning. Table 2 checks the estimated bits in advance to realize the bandwidth of the radio links that have been selected for appropriate LOCI and the CubeSats.

In the lunar mission, the visibility of the ground station in critical phases could be considered around 18 min at minimum; however, for a lunar orbit, visibility time would be large enough. S-band Up/Down link bit rate is 6 Mbps, X-band 400 Mbps, and Ka-band >400 Mbps, and, as can be seen in Table 2, the maximum downlink data volume is 1.6 GB. Therefore, X-band is an appropriate choice for direct transferring data to DSN. The mother ship utilizes high-gain X-band reflect array antenna for ground station transmission,

omini-direction S-band antenna for constellation link, and UHF for supporting links in any malfunctioning phase [47,48].

**Table 2.** Estimated Up/Down link data values.

| Uplink Data Estimation | |
| --- | --- |
| **Data** | **Value** |
| Telemetry | $160 \times 10^3$ |
| Software patch | $2 \times 10^6$ |
| Margin | $5.4 \times 10^5$ |
| Max data | $2.7 \times 10^6$ |
| Typical data | $1.4 \times 10^6$ |
| Downlink data estimation | |
| Payload data | $26 \times 10^7$ |
| Engineering data | $8 \times 10^3$ |
| Margin | $6.5 \times 10^7$ |
| Total downlink data per satellite | $33.6 \times 10^7$ |
| Total 5 satellites downlink data with mother ship | $168 \times 10^7$ |

The CubeSats have two back-to-back S-band antennas to be able to transmit payload and telemetry data without changing their orientation to the microsatellite. Omini-direction UHF is the supporting link in circumstances that the S-band transmitter is faced with malfunctioning (it could be considered a critical phase in a satellites scenario). Then the CubeSat transmits data through UHF transceiver; however, the data rate is very low, but less is better than nothing [49].

*4.5. Electrical Power Subsystem Design*

The majority of CubeSat projects calculate their power consumption according to simple Formula (1), which calculates the amount of power generation for a single solar cell:

$$p = \eta_n . S.A. \sin \phi \tag{1}$$

where p is power generated by a solar cell, S [W/m$^2$] sunlight power density, A is area of solar cell, $\eta_n$ solar cell efficiency, and $\phi$ is angle of sunlight irradiations with respect to the solar cell panel [50]. Since the project is for a lunar mission, the S is approximately 1415.54 W/m$^2$ with an average orbit time of 113 min on lunar orbit, according to [51], A is 3U CubeSat standard solar cells referenced from CubeSat shop company with $\eta_n$ efficiency 30%. A solar cell power generation is very dependent on the CubeSat attitude orientation. In [51], it was claimed that there are three different attitude orientation scenarios, free-orientation scenario, nadir-pointing scenario, and sun pointing scenario. Each of these orientations provides different power levels; for example, sun pointing generates maximum solar cells power capability, in free orientation, considering the average of solar cell power generation capability [51]. The system requires the satellites to always point to the Moon surface; therefore, the nadir-pointing scenario is our system attitude orientation scenario. The authors of [51] say that a 3U solar cell with nadir-pointing orientation approximately generates 5.84 W power. Table 3 estimates a CubeSat power budget. It helps to select the number of solar cells and their mechanism, either a body-mounted or deployable solar cells mechanism.

According to Table 3, body-mounted solar cells cannot provide sufficient maximum power consumption of the CubeSat; therefore, a deployable mechanism should be considered. To conclude, the functionality of Electrical Power Subsystem should be a deployable mechanism with 17.52 W power generation on average.

**Table 3.** CubeSat power budget.

| CubeSat Subsystem & Their Power Usage | | |
|---|---|---|
| Item | P(W) | Duty Cyclex/113 min |
| OBDH | 0.1 | 113/113 |
| UHF transceiver TX/RX | 4/0.48 | 5/113 |
| S-band transceiver TX/RX | 12/1.2 | 3/113 |
| ADCS MCU | 0.2 | 113/113 |
| Thruster | 21 | 1/113 |
| Sun sensor | 0.006 | 113/113 |
| Star tracker | 1 | 113/113 |
| IMU | 0.01 | 113/113 |
| THz Payload | 3.15 | 113/113 |
| | Power usage in scenario | |
| Mode | Active subsystems | power usage per mode (W) |
| safe | OBDH, ADCS, MCU, ADCS sensors | 1.616 |
| Recovery | OBDH, ADCS, MCU, ADCS sensors, Thruster | 2 |
| Normal | OBDH, ADCS, MCU, ADCS sensors, UHF RX/TX | 2.16 |
| Constellation | OBDH, ADCS, MCU, ADCS sensors, S-band, Thruster | 2.276 |
| Mode | Active subsystems | Power usage per mode (W) |
| Payload | OBDH, ADCS, MCU, ADCS sensors, THz | 5 |

### 4.6. Structure Design

The mass center of the CubeSats must stay within 2 cm measured from the XY plane in the z geometric direction. Since the structure protects the satellite from the harsh environment of space and the launch vehicle's vibration forces, some standards are required for designing a CubeSat structure. All deployable mechanisms, such as antenna and solar cells, must stay all together during launch and injection phases. Another factor that affects structure characteristics is mass budget: the mass budget is a priority that must be considered before designing the CubeSat structure. Table 4 shows the CubeSat mass budget. The first face inserted into P-POD is the –Z side. The project includes two principal bodies, LOCI and the 3U CubeSats. LOCI contains four P-PODs, a camera on the +X side, the nozzles of the thruster on –X side, two 3U deployed solar cells on +Y,-Y sides, and a reflect array on the +Z side. The CubeSat includes two 3U deployable solar cells on the ±Y side; under them, there are two S-band patch antennas to radiate omnidirectional radio signals. By using that layout, the CubeSats constellate without pointing orientation to LOCI, star tracker module is on +Z side, and THz spectrometer is positioned to the Moon surface, means on the–Z side (see Figure 8, CubeSat structure; Figure 9, LOCI structure). The dimensions of LOCI are 50 × 50 × 70 cm.

**Table 4.** CubeSat mass budget.

| Subsystem | | |
|---|---|---|
| Item | Mass (gr) | Quantity |
| Thruster | 500 | 1 |
| 3U solar cells | 150 | 3 |
| 20Whr battery and Hardwares | 350 | 1 |
| Main MCU | 100 | 1 |
| UHF transceiver | 75 | 1 |
| UHF ant | 50 | 1 |
| S-band transceiver | 132 | 1 |
| S-band ant | 75 | 2 |
| Star tracker | 250 | 1 |
| Structure | 304 | 1 |
| THz payload | 400 | 1 |
| Total mass | 2761 | |
| Margin | 239 | |

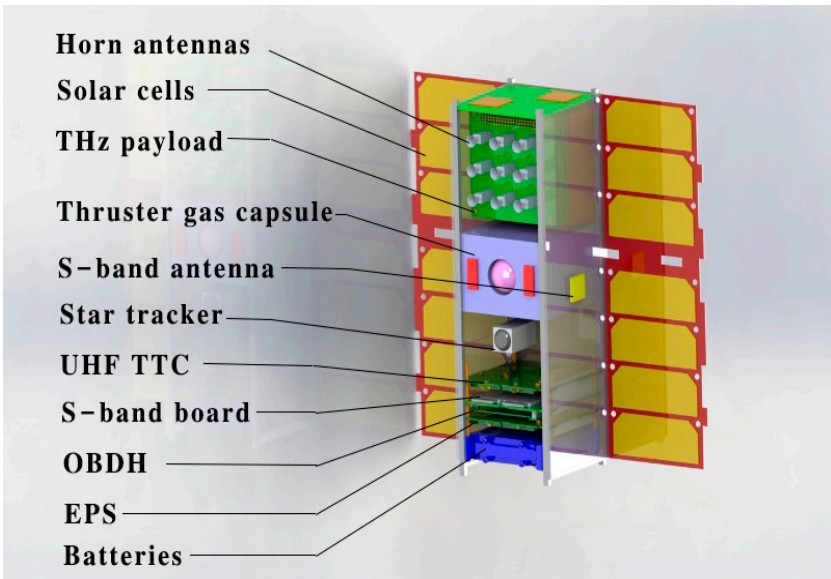

**Figure 8.** CubeSat characteristics.

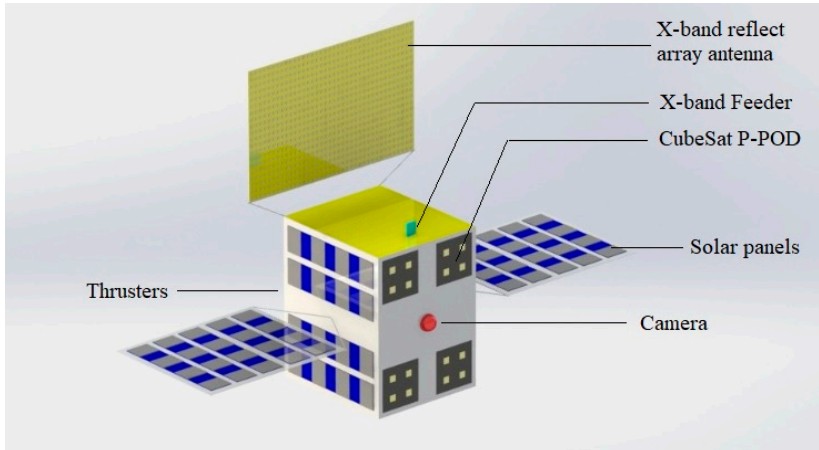

**Figure 9.** LOCI Microsatellite characteristics.

### 5. Conclusions

In this paper, we proposed an interplanetary mission with a cargo Microsatellite. We already know that there are missions such as MarCO (Mars Cube One), a 6U CubeSat targeting Mars, but a microsatellite has more electrical power, thruster fuel, and mass capacities; consequently, the mission can proceed farther into deep space and perform it in a shorter amount of time. A CubeSat with an electrical thruster reaches the Moon in approximately 900 days, and a microsatellite reaches it in 300. A microsatellite can transport a CubeSat into deep space, and the mission period could be done three times faster; in addition, the microsatellite could be developed with system characteristics such as a thruster and its fuel bank. Then, the microsatellite can use the gravity assist concept and travel to different planets and deliver CubeSats to each planet. For example, the microsatellite gest assistance from Earth's gravity and then goes toward Venus's orbit to deliver a CubeSat and then gets assistance from Venus's gravity and thrusts to Mars to deliver another CubeSat. All scenarios are possible with our proposed microsatellite platform with a small change in the system. On the other hand, the proposed 3U CubeSat is a platform for any interplanetary missions that it is required to be accomplished using a CubeSat. The constellation with five satellites gives us this opportunity to cover the Moon's surface monitoring two or three times per day, and the microsatellite with an x-band high data rate transmitter transfers the collected data to the Earth's ground station

with a small amount of time delay. Here, we proposed a kind of innovative THz heterodyne spectrometer for water identification. The payload has array horn antennas. The benefit of this layout is higher gain. We could have used an array patch antenna, but the maximum gain that can be reached using array patch antennas is approximately 5–6 dB, while that of a horn antenna is typically around 10–20 dB, which gives us better options to receive signals from the Moon's surface. To summarize, with water, we can complete interplanetary tours to prospect unknown planets, especially with artificial satellites as pioneers and CubeSats because of their cheap price and fast operation and ability to perfectly accomplish such missions. However, current CubeSat thruster technologies do not allow them to be fast-traveling objects, and this is the only weakness. Hence, we consider the microsatellite to transport the CubeSats, so that weakness is improved by the cargo microsatellite. The constellation provides more data in less time. We targeted water detection on the Moon because we believe the Moon is a window for the future of interplanetary missions. The Moon is one of the extraterrestrial natural satellites attracting scientists' attention to discover outer space. Spacecraft propellants refueling, plants growth, and oxygen for breathing all are possible with water. Water is life, and this scenario has been defined to find it.

**Author Contributions:** In this project, I have two colleages that assist me on this paper. Their contributions with me are as follows: Conceptualization by, V.R.; methodology by, V.R.; software by, V.R. and M.K.T.; validation by, W.H. and V.R.; formal analysis by, W.H.; investigation by, V.R. and M.K.T.; resources by, V.R.; data curation by, V.R. and W.H.; writing—original draft preparation by, V.R.; writing—review and editing by, V.R.; supervision by, Hu Weidnong; project administration by, V.R. and W.H.; funding acquisition by W.H. All authors have read and agreed to the published version of the manuscript.

**Funding:** This research received no external funding.

**Institutional Review Board Statement:** Not applicable.

**Informed Consent Statement:** Not applicable.

**Data Availability Statement:** The data that support the findings of this study are openly available in Yanghyo, K., et al at http://doi:10.1109/TTHZ.2019.2910988, reference number [36].

**Conflicts of Interest:** Vahid Rastinasab, Weidong Hu and Mohammad Kazem Tahmasebi declare that they have no conflict of interest.

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
