# Peer review of "Water Recognition on the Moon by Using THz Heterodyne-Spectrometer for Identifying the Appropriate Locations to Extract Water for Providing Oxygen for Breathing and Fuel for Spaceships’ Propulsion on the Moon with CubeSat"

_aerospace, doi:10.3390/aerospace8070186_

Round 1

Reviewer 1 Report

The topic is fine, the technical details coarse and below phase A level. The paper is a good brainstormer.

Author Response

Dear 

Greetings!

Thanks so much for your comments that you have written on my paper. I have revised the English grammar points, I share you my revised paper but also here I write you my modification, kindly please take a look and send me massage if it needs more modification.

line 33: modified to life support.

line 56: modified.

lines 60-61: Yeah, all waters on the moon are ice water, I edited and put a reference about the Moon's ice poles.

line 82: modified.

Figure 1&2&3&8: edited please take a look if that is not still clear.

line 171&172: edited, they were angles.

line 193-195: the sentence is modified, please check.

Figure 7: is deleted now.

line 279: edited to W symbol.

line 278: I have written a reference, inside the reference are the all analysis, because I did not want to duplicate the analysis inside my paper, therefore, I have only referenced the relative paper.

Table 3: is edited, please check.

line 321: the conclusion is modified, please check.

Thanks so much for all your efforts

I am looking for your reply.

Best Regards

Yours Sincerely 

Vahid

Reviewer 2 Report

The paper titled  "Water recognition on the Moon by using THz heterodyne-Spectrometer for identifying the appropriate locations to extract water for providing Oxygen for breathing and fuel for Spaceships’ propulsion on the moon with CubeSat"  proposes a constellation of CubeSats to search for water on the Moon. In the paper, the authors describe the rationale and the strategy to reach the proposed goal. It touches on a theme of actuality. Nevertheless, I think that in this shape the paper could not be published.
The paper gives the impression that it is written in hurry and submitted in its draft state. Most of it is not understandable due to poor English both for word choice and construction. Obviously, it was not a yardstick on the validity of the ideas described in the paper, but anyway it was an obstacle to the understanding of their validity.

I try to outline some main concerns:

General:
- Bibliography: most of the bibliography lacks clear identification. Sometimes only the name of the workshop is reported with any pieces of information about the volume or the page.
- there is no final list with an explanation of the used Acronyms. 
- Pictures in Figures 1, 2, 3, and 4 are too small, and not easy to read. 
- Figure 8 is a really low-resolution picture. It is not understandable.
- Most of the Figures are not referenced in the text.
- English has not been revised. Most sentences are not clear and hard to understand. I began to indicate someone to the author but then I gave-up
- There are a lot of words separated by a hyphen like e.g. drink-ing ...

Major:
The introduction section is important to introduce the topics of the paper giving the right references to allow the not expert reader to enter the mat. This is not the case here. It gives some hints but does not explain the status of this particular activity of the solar system exploration, the proposed mission, the problems related to a lunar station just to mention some topics touched in the introduction.

Other particular major points: 
- Row 23: Why you are speaking of asteroids in the Galaxy? The asteroids in the Solar System are not enough?
-Row 31 because the moon is the nearest asteroid to --> This sentence contains a concept that is completely wrong. In a scientific paper, but also in an outreach type paper, you cannot make such parallelism. The Moon is a satellite and cannot be compared to an asteroid for a lot of reasons! E.g. formation and evolution just to mention any. Please remove it.
- The dimensions of LOCI are missing.
- what kind of launcher will be used to bring the LOCI into the lunar orbit?

Minor:
Row 24: What kind of precious materials?
Row 27: Reference 1: please put a clear reference. There are just the authors and the title. What is it? is this a proceeding of a workshop? a paper on a Journal?
Row 27 - 29: this sentence is not so clear, please rephrase it.
Row 42: TROPICS, MicroMAS, and MiRaTA are The Most recent THz remote --> TROPICS, MicroMAS and MiRaTA are the most recent THz remote...
Row 46 that we will be disusing about it in the paper to vast that scope for this idea.--> This sentence is completely understandable ... please check it!
Row 59 ..icy water on the moon’s poles --> Put a reference here!
Row 78-79 opportunity to vast CubeSat for --> to vast? it is not an English verb ...perhaps the authors would like to use it to cast?
Row 118 -125: what is the meaning of these sentences? ... Who are the authors of which the authors are speaking about?
Row 122: The water vapor is decomposed with hydrogen and goes into deep space ... Decomposed? in which way? this sentence is not clear. Who is going into deep space? Water vapor? Hydrogen?

Row 135 up to 139: it seems to me that all the spatial missions comprise those subsystems ...

Row 148 - 149 you are using for the first time acronyms not explained. 
Row 153 three 10*10 cm PCBs with 3.15W power--> PCB ?
Row 170 to 173: This sentence is not clear. Please rephrase it... there are also not clear symbols.
Row 212 ADCS ... that means?
Row 221 - 222 The sentence is unclear ...
Row 238 TTC ... that means?

Author Response

Dear 

Greetings!

Thanks for all your valuable comments, I have revised the paper with all grammatical errors, pleas take a look and kindly reply me. Farther I am answering your other comments.

General:

-The references are modified. The introduction is modified with more related papers. I have tried to survey on pervious works, similar to literature review.

-I have written all acronyms in paper content.

-Figures 1&2&3&4&8 are revised.

-English is revised and about hyphens, that belongs to the template, any sentence has a limitation on its total words so it connects end of  the line words by a hyphen.

Major:

I have modified the introduction with more relative papers and survey on pervious researches.

Other particular major points:

Row 23: modified to solar system.

Row 31: Modified.

Dimensions of LOCI is 50*50*70cm.

LOCI uses Aerjet MPS-130 thruster to travel moon.

Minor:

Row 24: precious materials such as platinum, gold, iridium etc.

Row 27: Reference 1, that is a conference paper I wrote it, apologize. 

Row 42: modified.

Row 46: Modified.

Row 59: Reference is putted.

Row 78-79: The verb is modified.

Row 118-125: all are LOCI and the CubeSats scenarios, and the author who is saying those sentences is me.

Row 122: The sentence is revised, Hydrogen and oxygen go into deep space.

Row 135-139: The only different that I realized between commercial and outer space CubeSats was in their ADCS systems where magnetic based sensors and actuators are useless in outer space, therefore, it might look similar other commercial CubeSats.

Row 135-139: Modified.

Row 153: I meant the electrical board power consummation. The sentence is revised.

Row 170-173: modified.

Row 212: ADCS means Attitude Determination and Control System. I also put in the paper.

Row 221-222: The sentence is modified. Please check.

Row 238: TTC means Telemetry Tracking and Command system.

Thanks so much for your efforts.

I am looking for your reply.

Best Regards 

Yours sincerely 

Vahid Rastinasab

Round 2

Reviewer 2 Report

The paper had a noticeable improvement since the first version.
There is still a disorder in the references. If you use the number notation, you have to follow the appearance order. Just to say, Reference 55 could not be cited before reference 3. Furthermore, I have individuated some miss print and the minor concerns listed in the following:

Minor
- 1. Introduction
- reference n. 3 comes before reference n. 2
- pag. 1 for example on the moon,--> for example on the Moon, Please, correct all along with the paper
- Reference 6 is not recalled in the text ...
- Reference 17 is not recalled in the text ...

3. Lunar Orbit CubeSat Injector Scenario
-After Attitude Determination System (ADS) rate<???°  --> It is quite unusual to see in a paper TBD instead of a number. Could you please indicate an approximate value? if not, please modify the sentence in a way that TBD will be not written and describe why the ADS rate could not be defined. 

4. System level design
- Part of this paper includes a cargo Microsatellite (named LOCI)--> Part of this paper includes a description of a cargo Microsatellite (named LOCI)
- Figure 3 is not cited in the text.

Author Response

Dear 

I have revised based on your comments. Please receive the file as it is attached via this email.

Thanks so much for your efforts

Best Regards 

yours sincerely

Vahid Rastinasab
